# Your Model Is Not Neutral—It's Just Well-Socialized

## Abstract

We argue that contemporary instruction-following language models are best understood not as "neutral" calculators over text but as *well-socialized* systems: they internalize, reweight, and strategically present culturally dominant norms and rules. We synthesize evidence spanning pretraining corpora composition, alignment pipelines (RLHF/RLAIF), and evaluation artifacts, and we propose a measurement protocol that distinguishes *norm adherence* from *value neutrality*. Our position reframes bias mitigation: the question is not whether models encode values, but *which* values they surface, to whom, and under what prompting regimes. Our contribution is conceptual and procedural: a vocabulary and checklist, not new metrics or experiments.

## 1 Introduction

LLMs are trained on vast web corpora that imprint human regularities, including stereotypes and social hierarchies [7, 9]. Subsequent scaling and deployment has foregrounded risks and normative choices [2, 4]. Alignment methods such as RLHF [20] and RLAIF/Constitutional AI [1] further *curate* model behavior toward socially sanctioned responses. The resulting systems often *exhibit* politeness, safety, and fairness while still inheriting corpus-level asymmetries. We label this phenomenon *well-socialization*: optimization that shapes surface behavior to conform to desired norms without rendering the representation space norm-neutral.

**Claim.** LLMs are not neutral; they are *socialized* by (i) data, (ii) objective and feedback design, and (iii) evaluation/benchmarking conventions. Recognizing this clarifies why debiasing sometimes improves benchmark scores yet leaves downstream disparities intact.

## 2 From Neutrality to Well-Socialization

**Pretraining socialization.** Web-scale corpora (e.g., C4) include skewed topical and demographic coverage and blocklist filtering that *systematically* drops minority varieties [9]. Word- and sentence-level associations capture culturally learned stereotypes [7]. These are not incidental noise but stable statistical structure.

**Alignment socialization.** Instruction tuning with human feedback translates community norms into loss gradients [20]. Constitutional AI replaces or supplements humans with principles, still codifying a normative charter [1]. In both regimes, optimization encourages *performative* adherence to rules (e.g., refusal templates), which can mask representational biases.

**Documentation socialization.** Model and dataset documentation (model cards and datasheets) make norms explicit and can redirect optimization targets [17, 10]. This is constructive socialization: it steers practice toward transparency and stakeholder considerations.

Submitted to 39th Conference on Neural Information Processing Systems (NeurIPS 2025). Do not distribute.

## 3 Evidence: Values In, Values Out

**Corpus effects.** C4 analysis shows content source skew and blocklist-induced deletions affecting marginalized dialects [9]. At the representation layer, canonical results (WEAT) exhibit human-like biases learned from unlabeled text [7].

**Behavior under bias probes.** CrowS-Pairs and StereoSet demonstrate persistent stereotypical associations in strong LMs [19, 18]. In toxicity detection pipelines, dialectal bias produces higher false positives on AAE [23]; analogous mechanisms can arise in generative toxicity proxies.

**Alignment trade-offs.** RLHF improves helpfulness and reduces overt toxicity [20] while leaving latent associations that reappear under adversarial or distribution-shifted prompts. Constitutional AI encodes a principled rulebook but *selects* which rules govern surface behavior [1]. Thus, alignment changes *which* values are expressed, not whether values are present.

**Scholarly meta-evidence.** Surveys emphasize that "bias" work is intrinsically normative and must ground harms and stakeholders [4]. FAccT meta-analyses document how ML research itself encodes community values [3]. Broader STS analyses trace datasets, labor, and resource externalities [8]. We operationalize well-socialization along three measurable axes, adding concrete metrics and multiple corroborating references for each.

## 4 Operationalizing Well-Socialization

**Representational Associations.** (i) Compute classic WEAT/SEAT scores on base and aligned LMs using [7] and [16] templates; for contextual encoders or LLMs, use masked/probe variants from [14]. (ii) Audit residual bias after debiasing or alignment ("lipstick on a pig") by replicating [12] and inspecting *indirect* bias in geometry. (iii) Test word-level association shifts versus sentence-level behavior by comparing pre-/post-alignment embeddings [6] and generation on stereotype prompts.

**Surface Norm Adherence.** (i) Quantify policy conformance and refusal propensities under instruction-following regimes using preference-optimized models [25, 20] and constitutions [1]. Report refusal rate, helpfulness/harmlessness Elo, and policy-consistency. (ii) Stress-test with toxicity and safety suites (toxicity rate, conditional toxicity, detox controllability) using RealToxicityPrompts and multi-metric dashboards like HELM [11, 15]. (iii) Probe brittleness via automated red teaming and universal jailbreak suffixes; measure attack success rate (ASR) and robustness under prompt perturbations [22, 26].

**Context Sensitivity.** (i) Dialect and register audits: evaluate toxicity and moderation outcomes on African-American English (AAE) and other varieties [5, 23]. (ii) Cross-modality spillover: check text behavior conditioned on ASR transcripts from demographically diverse speakers (error-driven bias) [13]. (iii) Identity-controlled templates: use HOLISTICBIAS descriptors across demographic axes and QA bias benchmarks like BBQ to measure identity-conditioned preference shifts, refusal asymmetries, and harms [24, 21].

A model is "well-socialized" if surface norm adherence is high while representational associations remains non-neutral or context sensitivity reveals conditional disparities. Reporting should include all three, with stratified slices and uncertainty.

## 5 Implications

**Benchmarking.** Include paired probes: (A) value expression under neutral prompts; (B) value expression under adversarial/realistic prompts; (C) robustness across sociolinguistic varieties [19, 23].

**Design.** Treat alignment as *policy learning*. Make the normative charter explicit (constitution/model card) [17, 10]. Audit feedback provider pools; diversify principles and rater demographics.

**Governance.** Document data lineage and filtering, publish datasheets/model cards with stakeholder input, and report failure cases and trade-offs rather than implying neutrality [10, 17].

## 6 Conclusion

Neutrality is not the default; socialization is. LLMs learn cultural regularities from pretraining and then are coached to present rule-following behaviors via alignment. Recognizing this reframes fairness from "removing bias" to *governing which values are encoded and expressed*, with transparent charters, pluralistic feedback, and multi-layered evaluation.

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
