# OpenReview forum: "Your Model Is Not Neutral—It's Just Well-Socialized"
_EurIPS.cc/2025/Workshop/UPLB — UPLB2025_

### Official Review · Reviewer_ZioV · 2025-10-27
**The authors of this work claim that LLMs encode intrinsic biases which are usually not eliminated in the training or downstream process, but rather are corrected upstream making these models not really neutral, but just "well-socialized".**

**Rating:** 3
**Confidence:** 2

**Review:**

Besides the abstract and the brief introduction, I really struggled to understand the purpose of this paper. If I have well understood the main thesis, i.e. that LLMs biases are not tackles downstream but rather upstream, I find it difficult to see if this is an original claim. The paper itself is not well structured, and lacks of clarity. It is divided in many sub-paragraphs, which seem just a collection of bullets points which are difficult to link together. Moreover, many connections to previous works are not made or are unclear, missing many times the definition of the abbreviation: for instance, RLHF (lines 5 and 20). I suggest to the authors to re-write the article, putting an effort to structure it in a concise, yet logically sound way.

---

### Decision · Program_Chairs · 2025-11-03

Accept (Poster)